# OpenReview forum: "Non-Commutative Spectral Geometry for Adaptive Quantum-Classical Drug-Target Interaction Prediction"
_ICLR.cc/2025/Conference — Submitted to ICLR 2025_

### Official Review · Reviewer_24YN · 2024-10-23

**Soundness:** 1
**Presentation:** 1
**Contribution:** 2
**Rating:** 3
**Confidence:** 3

**Summary:**

In this article, the authors propose Non-Commutative Geometric Adaptation for Molecular Interactions (NCGAMI) for drug-target interaction (DTI) prediction. They claim that their model is a groundbreaking unified framework that integrates advanced concepts from non-commutative geometry, optimal transport theory, and quantum information science, and their results are significantly outperforms state-of-the-art models.

**Strengths:**

It is interesting to see that the authors try to bridge the gaps between advanced mathematics, AI models and applications.

**Weaknesses:**

1) The presentation of the paper is very bad! The authors simply present the general theorems and definitions in information geometry, algebraic topology, and quantum information and claim they proposed "a unified framework". The connections are totally missing.
2) The authors made too many overclaims in their paper. For instance, 1) their works are "groundbreaking" framework; 2) Their works reveal "fundamental connections" to "deepest areas of mathematical and theoretical physics", etc.
3) Clear mistakes about the statements of GNN models. They claim that GNNs "are fundamentally limited by their adherence to classical
probability theory and Euclidean geometry" and "drawback of GNNs is their focus on local neighborhood nodes, potentially overlooking the comprehensive global three-dimensional structures and edge information".

**Questions:**

1) It is totally unclear how the proposed algorithms are incorporated into their deep learning architecture (of NCGAMI). In fact, the authors do not give any detailed explanation of the NCGAMI models, and how their theoretical framework is reflected or incorporated into their NCGAMI models. What is the key difference with existing similar architectures? Where are the improved parts?
2) There is huge gap between the proposed theoretical framework and real applications. For instance, why we should consider "non-commutative" geometry? How the "topological space of molecular configurations" in related to DTI? What is the biological meaning of "Pharmacological sheaf cohomology"?
3) The authors lack a good understanding of GNNs. It is NOT true that "Traditional machine learning approaches to DTI prediction have primarily relied on classical statistical methods and, more recently, on graph neural networks (GNNs) and attention mechanisms. While
these methods have shown promise, they are fundamentally limited by their adherence to classical probability theory and Euclidean geometry". GNNs are designed for non-Euclidean geometry!  Further, "Nevertheless, a drawback of GNNs is their focus on local neighborhood nodes, potentially overlooking the comprehensive global three-dimensional structures and edge information". There are many GNN models that consider edge information and global structure information.
4) The authors only compare their model with the relatively simple models, such as "DeepDTA". Many important GNNs are not considered, for instance Graph transformer models and graph attention models. The compared models are clearly not "state-of-the-art".

---

### Official Review · Reviewer_Nid2 · 2024-10-31

**Soundness:** 2
**Presentation:** 2
**Contribution:** 2
**Rating:** 5
**Confidence:** 3

**Summary:**

To deal with the drug-target interaction (DTI) tasks, the authors develop a unified non-commutative geometric framework based on Riemannian geometry, algebraic topology, category theory, sheaf theory, etc, providing a deep connection between the algebraic structure of DTI prediction and the geometry of optimal transport.

**Strengths:**

1. The study utilizes various cutting-edge math techniques to address the DTI problem, this is novel.

2. In theoretical discussions such as Theorem 3.3, and 3.4, the authors use proof sketches, which makes the content more accessible to readers.

3. The authors provide the code, which helps the reproduction of the results.

**Weaknesses:**

1. The (Unsupervised) Domain Adaptation is a long-standing research domain and numerous papers have been published, even if we narrow it down to DTI [1].  Unfortunately, this study hardly reviews this profound and broad research domain[2,3], which makes the proposed method poorly motivated.

[1] Bai P, Miljković F, John B, et al. Interpretable bilinear attention network with domain adaptation improves drug–target prediction[J]. Nature Machine Intelligence, 2023, 5(2): 126-136.

[2] Wilson G, Cook D J. A survey of unsupervised deep domain adaptation[J]. ACM Transactions on Intelligent Systems and Technology (TIST), 2020, 11(5): 1-46.

[3] Wang M, Deng W. Deep visual domain adaptation: A survey[J]. Neurocomputing, 2018, 312: 135-153.

2. I feel the disconnection between the introduction of non-commutative geometric framework and its application to deep learning models. After all,  I saw the use of GCN, KAN and Mamba models, I am not sure how the theoretical framework helps to better design a neural network architecture or loss function.

**Questions:**

1. Following the above, could the author tell me how the theoretical framework contributes to your model design or experimental tasks?

---

### Official Review · Reviewer_a3Yy · 2024-11-01

**Soundness:** 2
**Presentation:** 1
**Contribution:** 1
**Rating:** 1
**Confidence:** 3

**Summary:**

The authors present the Non-Commutative Geometric Adaptation for Molecular Interactions, which is a novel Unsupervised Domain Adaptation (UDA) framework, and applied it in DTI. To this end, they first proposed the concept of UDA in the field of DTI with corresponding optimization method, and then expands the concept and method in terms of sympletic geometry (section 4) non-commutative geometry (section 5). To introduce these concepts, they fused various domains' complex concepts, such as quantum information theory or algebraic topology.

Though theory may seems to be soundful, this manuscript is not sufficient enough for ICLR publication. There's no detailed preliminaries about theory, related works, and proofs of theorems. Furthermore, the performance improvement is very marginal, and the baselines for experiments are seems to be miss-selected to demonstrate the effectiveness of proposed framework.

**Strengths:**

Proposed novel approach in drug-target interaction, which utilizes the concept of unsupervised domain adaptation.

**Weaknesses:**

### W1. Unkind manuscript for general audience

**W1-1. Lack of information about preliminaries**

As a researcher in protein-ligand interaction domain, I strongly believe that the terms like cohomology and time-dependent Hamiltonian may be difficult to understand in general for the readers for this field. However, the manuscript does not provide any references to start with nor introductory explanation of these concepts. This will limit the impact of the work, regardless of the novelty and strength of the work.

**W1-2. Lack of information about related works**

As far as I understand, the main topic of the manuscript is development of theoretical framework on unsupervised domain adaptation, but the related works on UDA or its theory are not provided. Since related works inform about baseline models or theory, lack of related works in this part also limit the impact of the work, regardless of the novelty and strength of the work.

**W1-3. Lack of detailed proofs**

The authors proposed several theorems and corresponding proofs. However, almost *every proof is explained with sequential steps, which are still not very trivial*. For example, the second step of the proof of theorem 3.4, the authors stated that "Verify the conditions for consistency of M-estimators in the presence of nuisance parameters (transport map)". However, this seems not trivial. If page limit matters, the detailed proofs for each theorem can be provided in the appendix.

**W1-4. Lack of reference**

For several proofs and texts, there's no reference provided. For example, in the third step in the proof of theorem 3.4, Huber-Donsker-Varadhan theorem should be cited properly. (I couldn't find the proper citation for this) There are more cases (not all) that needs reference:
- Jarynski equality[1]
- Crooks fluctuation theorem [2]
- Tomita-Takesaki modular theory [3,4]
- see Q4 (About limitations of current works in introduction)

### W2. Unclear explanation for applying theory in practical application

Although practical application of proposed framework based on theory is not very trivial, authors do not provide any explanation for implementation. For example, the term "UDA implicit data augmentation" is shown first-time and last-time in the ablation study (section 6.4). The authors **should** clarify the detailed implementation steps in the manuscript.

### W3. Not well-represented results (see Q2)
**W3-1. Results do not seems to demonstrate the effectiveness of theory in terms of UDA and DTI**

1. **Performance evaluation in terms of UDA**: Since the main idea of this work is UDA framework for DTI, to demonstrate the effectiveness of *theoretical framework itself*, authors should compare their framework with other UDA frameworks, while applying them to DTI. If this is not possible, the reason for this should be clearly explained.
2. **Performance evaluation in terms of DTI**: To demonstrate the effectiveness of the proposed theory in DTI, it is essential to show that it outperforms existing methods on this task. The results do not exhibit significant performance improvements over other classification models, which raises concerns about the theory’s effectiveness. Additionally, the manuscript lacks an analysis of the underlying reasons for the observed lower performance compared to baseline models, further questioning the robustness of the theoretical contributions. Moreover, presenting the comparative performance metrics in a table format rather than a graph would allow for more intuitive and straightforward comparisons across different models.

**W3-2. Lack of comparison with other methods or more datasets**

Currently, results in the manuscript, comparing with DTI models, is not very comprehensive in two aspects: baseline models and datasets.
1. **Baseline models:** The benchmark should be done with more recent models[5,6,7], that outperform currently compared baselines. If there's any reason that cannot benchmark explicitly with those models, the authors should explicitly note them.
2. **Datasets:** To better demonstrate the robustness of proposed framework, the authors may consider benchmarking along more dataset, such as BindingDB[8], BioSNAP[9].

## References
[1] Jarzynski, Christopher. "Nonequilibrium equality for free energy differences." Physical Review Letters 78.14 (1997): 2690.

[2] Crooks, Gavin E. "Entropy production fluctuation theorem and the nonequilibrium work relation for free energy differences." Physical Review E 60.3 (1999): 2721.

[3] Takesaki, Masamichi. Theory of operator algebras II. Vol. 125. Berlin: Springer, 2003.

[4] Summers, Stephen J. "Tomita-Takesaki modular theory." arXiv preprint math-ph/0511034 (2005).

[5] Peng, Lihong, et al. "BINDTI: a bi-directional intention network for drug-target interaction identification based on attention mechanisms." IEEE Journal of Biomedical and Health Informatics (2024).

[6] Cheng, Zhongjian, et al. "IIFDTI: predicting drug–target interactions through interactive and independent features based on attention mechanism." Bioinformatics 38.17 (2022): 4153-4161.

[7] Ye, Yuqing, et al. "PHCDTI: A multichannel parallel high-order feature crossover model for DTIs prediction." Expert Systems with Applications 256 (2024): 124873.

[8] Gilson, Michael K., et al. "BindingDB in 2015: a public database for medicinal chemistry, computational chemistry and systems pharmacology." Nucleic acids research 44.D1 (2016): D1045-D1053.

[9] Zitnik, Marinka, Rok Sosic, and Jure Leskovec. "BioSNAP Datasets: Stanford biomedical network dataset collection." Note: http://snap. stanford. edu/biodata Cited by 5.1 (2018).

**Questions:**

**Q1. Applicability of UDA Framework Beyond Drug-Target Interaction**

In my opinion, the proposed method, which presents theoretical framework of UDA, can be applied in various domain since the concept of *pharmacological* sheaf (and the other definitions about pharmacophores) can also be applied in other domain as *domain-specific* sheaf. Image and graph domain already have many literatures.[1] Is there any reason that authors applied the proposed method specifically for drug-target interaction domain?

### Q2. Questions about results
**Q2-1. Baseline Training Details and Fairness of Comparisons**

About experimental settings for baselines, how did authors trained baseline models to compare with the model trained with NCGAMI? The baseline models do not use UDA frameworks, and if they are only trained on source data, it is not fair comparison. Please clarify how the baseline models are trained.

**Q2-2. Demonstrate Domain Adaptation to Data-Deficient Protein Families**

For the practical aspect, I've imagine applying UDA for data-deficient area in protein-ligand interaction, such as relatively small protein family.
If the authors can show the proposed framework's domain transfer in terms of training source data in GPCR(large, relatively data-sufficient protein family) and applying it more smaller data-deficient protein family will show NCGAMI's effectiveness better.

**Q2-3. Appropriateness of Random Dataset Splitting for Domain Adaptation**

Is random splitting of dataset is suitable to prove the performance of domain adaptation? In my opinion, simply using random splitting may lead to the same probability measures $\mathcal{P}_s=\mathcal{P}_t$, which directly contrasts the fundamental assumption of domain adaptation problem $\mathcal{P}_s\ne\mathcal{P}_t$, as introduced in section 3.

**Q3. Concept Figure**

The concept figure does not effectively illustrate the proposed framework. Instead of detailing the specifics of NCGAMI, the figure appears to represent a generic neural network architecture for DTI prediction, thereby omitting the UDA component. Additionally, the abbreviation “SSM” is unclear—does it stand for state-space model? The manuscript should ensure that all abbreviations are defined and that the concept figure accurately reflects all key components of the proposed framework, including the UDA.

 **Q4. References in Introduction**

In the second paragraph of introduction, is there any reference for the following? "These limitations become particularly apparent when attempting to model the quantum mechanical aspects of molecular interactions or when dealing with the high-dimensional, non-Euclidean spaces characteristic of chemical compound libraries and protein structures."

**Q5. Consistency between Objective and Hamiltonian**

In Theorem 4.3, the authors aim to preserve important domain-invariant features, which sounds good.
However, there appears to be a potential inconsistency regarding the treatment of the overall objective $\mathcal{L}_\text{unified}$.
As I understand, the objective is intended to be minimized during training.

Concurrently, the manuscript associates $\mathcal{L}_\text{unified}$ with the Hamiltonian $H$.
In classical Hamiltonian dynamics, $H$ is conserved over time, meaning it remains constant and is not subject to minimization.

This raises the question: *How do the authors reconcile the minimization of $\mathcal{L}_\text{unified}$ with the conservation properties of a Hamiltonian system?* If $\mathcal{L}_\text{unified}$ serves as the Hamiltonian, minimizing it would contradict the inherent property of $H$ being conserved. Clarification on this point would enhance the theoretical coherence of the manuscript.

**Q6. Ablation study**

Is there any reason for conducting ablation study on model architectures, even though it is not mentioned in the manuscript?

**Q7. (Very minor, just out of curiosity)**

Why did the authors select *lifelong learning* in the primary area?

## References
[1] Liu, Xiaofeng, et al. "Deep unsupervised domain adaptation: A review of recent advances and perspectives." APSIPA Transactions on Signal and Information Processing 11.1 (2022).

---

### Official Review · Reviewer_geQy · 2024-11-03

**Soundness:** 1
**Presentation:** 1
**Contribution:** 2
**Rating:** 3
**Confidence:** 3

**Summary:**

This paper presents NCGAMI (Non-Commutative Geometric Adaptation for Molecular Interactions), a novel framework for drug-target interaction prediction that integrates non-commutative geometry, optimal transport theory, and quantum information science. Their approach demonstrates incremental performance gains over baselines when evaluated on the Human (Liu et al) and DrugBank datasets.

**Strengths:**

- Solid theoretical foundation merging non-commutative geometry with pharmacological applications.
- Clear connection between statistical mechanics and domain adaptation via fluctuation theorems

**Weaknesses:**

- Excessive mathematical complexity without clear practical benefits
- The quantum adiabatic optimization algorithm's practical implementation details are unclear
- The performance improvements (1.01-2.34% AUC) are relatively modest given the theoretical complexity

**Questions:**

- What is the computational complexity of implementing the sheaf cohomology calculations?
- How does the model handle numerical instabilities in the quantum adiabatic optimization?

---

### Meta-Review · Area_Chair_ch3m · 2024-12-12

**Metareview:**

This submission presents a framework for predicting drug-target interactions. To this end, several concepts from non-commutative geometry, optimal transport theory, and quantum information science are drawn together. While the main merit of the paper lies in a somewhat unconventional setup that brings together new areas, it is suffering from presentation issues and a weak experimental setup. This makes it hard to assess the utility of the proposed method and understand its contextualisation. As one concrete example of an experimental shortcoming, none of the experimental results is described with standard deviations and in tabular format, precluding a simple comparison with other methods. Given these issues, I believe that another round of reviews would be warranted, which would constitute a major revision of the work. This is not feasible during the conference cycle, so I suggest _rejecting_ the paper for now.

I want to raise one unfortunate aspect, namely the lack of a rebuttal by the authors as well as the general lack of communication. Some of the points by the reviewers could have been addressed even within the reviewing period, giving the submission potentially a better chance of acceptance. It is unfortunate that the authors did not make use of that opportunity, and I sincerely hope that they will take the comments by reviewers into account when improving their work.

**Additional Comments On Reviewer Discussion:**

A discussion was not started by the authors, leaving the unanimously negative reviews unanswered. Reviewers raised predominantly three points of concern, viz:

1. Substantial presentation issues
2. Problems with the experimental setup
3. A missing contextualisation of the results (i.e. given the fact that the performance gains appear to be relatively _modest_, a proper discussion of the pros and cons of the proposed method would have been welcome)

I find myself in agreement with these points and since no author response was provided, I have to judge the paper based on the initial reviews and my own assessment.

---

### Decision · Program_Chairs · 2025-01-22

Reject